# Cytokine Profile at Diagnosis Affecting Trough Concentration of Infliximab in Pediatric Crohn’s Disease

**DOI:** 10.3390/biomedicines10102372

**Published:** 2022-09-23

**Authors:** Yiyoung Kwon, Eun-Sil Kim, Yoon-Zi Kim, Yon-Ho Choe, Mi-Jin Kim

**Affiliations:** 1Department of Pediatrics, Samsung Medical Center, Sungkyunkwan University School of Medicine, Seoul 06351, Korea; 2Department of Pediatrics, Kangbuk Samsung Medical Center, Sungkyunkwan University School of Medicine, Seoul 06351, Korea

**Keywords:** Crohn’s disease, children, TNF-α, infliximab, cytokines, T cell

## Abstract

Background: This study aims to measure the concentration of cytokines produced during the inflammation process to investigate if there are any differences in response to treatment of pediatric Crohn’s disease and to determine if the initial tumor necrosis factor-alpha (TNF-α) level affected the trough concentration of infliximab (IFX). Methods: This study included 30 pediatric patients with moderate-to-severe Crohn’s disease. At the time of diagnosis, blood samples were collected for the measurement of cytokines (IL-6, TNF-α, IL-17A, and IL-10). Blood samples were extracted from patients who had begun IFX treatment to measure the IFX trough concentration immediately before the fourth dose administration. Results: All cytokines (TNF-α, IL-6, IL-10, and IL-17A) were significantly higher in patients who did not achieve clinical or biochemical remission than in those who did (*p* = 0.027, 0.006, 0.017, 0.032, respectively). TNF-α had a negative correlation with the IFX trough concentration (Pearson coefficient = −0.425, *p* = 0.034). The diagnostic capability of the initial TNF-α concentration to predict under the therapeutic IFX trough concentration, defined as less than 3 µg/mL, had an area under the receiver operating characteristic of 0.730 (*p* = 0.049). The TNF-α concentration was set at 27.6 pg/mL as the cutoff value. Conclusions: Measuring cytokines at the time of diagnosis can be used to predict the treatment response. Measuring the initial TNF-α concentration may help to predict the treatment response to IFX. When the initial TNF-α concentration is greater than 27.6 pg/mL, a higher dose of IFX may be more appropriate than routinely administering 5 mg/kg of IFX to maintain the therapeutic concentration.

## 1. Introduction

Crohn’s disease (CD) is a chronic inflammatory bowel disease (IBD) whose etiology is uncertain [1]; however, immune system dysfunction may be a contributing factor [2]. The mucosal adaptive inflammatory immune response in the gut activates T cells classified as T helper cell (Th) 1, Th2, Th 17, and T regulatory (Treg) pathways through macrophage and dendritic cell [3,4,5]. Figure 1 is a simple schematic illustration showing the inflammation process in IBD drawn by the authors.

Consequently, numerous studies on the cytokines secreted by macrophage and each T cell subset were conducted, and tumor necrosis factor-alpha (TNF-α) is an important factor promoting chemokine secretion, epithelial barrier destruction, and epithelial cell apoptosis [6]. As it became clear, biological agents with anti-TNF-α roles, such as infliximab (IFX) and adalimumab, were used as treatment. Nonetheless, 10–30% of IBD patients treated with anti-TNF-α do not respond to the treatment [7]. Moreover, some patients lose their response to anti-TNF-α later on because of immune tolerance or antibody formation [8]. Consequently, research on other different cytokines is being actively conducted in CD.

The activation of T cells and Th17 pathways are also known to play a role in the pathogenesis of CD [9]. Interleukin (IL)-6 is a cytokine that is secreted by macrophages and is thought to be important in the process of activating T cells [10]. Th17 cells are abundant in the intestine, primarily in the terminal ileum, and secrete IL-17A and other cytokines [11,12]. Several studies have found that IL-17A mediates proinflammatory functions, such as the secretion of matrix metalloproteinases, which result in intestinal fibrosis [13]. A lack of regulatory cytokines secreted by Tregs can also lead to the onset of CD. IL-10 is a representative cytokine of regulatory cytokines that inhibits both antigen presentation and proinflammatory cytokine release [14]. Defects in the IL-10 and IL-10 receptors have been described in early-onset IBD [15,16].

Although many studies on these cytokines have been conducted, there are still limitations to their clinical use and therapeutic applications due to the complexity of the immunologic pathways and limited clinical data. Consequently, this study, as clinical data on cytokines in IBD aims to measure the concentration of cytokines produced during the inflammation process to evaluate the differences in responses to IFX treatment. This study also aims to see what effect initial TNF-α levels had on IFX trough concentrations.

## 2. Materials and Methods

### 2.1. Patients and Study Design

From June 2020 to June 2021, 30 pediatric patients (<19 years) newly diagnosed with moderate-to-severe CD were enrolled in this study. CD was diagnosed in accordance with the European Society for Pediatric Gastroenterology, Hepatology, and Nutrition (the Porto criteria) [17]. All patients underwent a diagnostic process that included laboratory and stool calprotectin testing, endoscopic evaluation (both esophagogastroduodenoscopy and colonoscopy), and magnetic resonance enteroclysis.

After an endoscopic examination confirmed that the patient had moderate-to-severe CD, blood samples were drawn for cytokine measurement. All patients began treatment with mesalazine and immunosuppressants (azathioprine or methotrexate). Corticosteroids were never used to treat any of the patients. For 8 weeks, some patients received dietary therapy with exclusive enteral nutrition (EEN). IFX intravenous therapy was started in patients who did not improve with oral drug treatment, showed minimal improvement, or improved and then worsened again. Therefore, IFX treatment was started approximately 8 to 12 weeks after diagnosis.

IFX maintenance treatment was administered at 8 week intervals following induction treatment, which was scheduled to be administered three times at weeks 0, 2, and 6. Colonoscopy and stool calprotectin testing were performed on patients receiving IFX treatment at the time of the fourth dose. A colonoscopy and stool calprotectin evaluation were performed 4 months after the start of treatment for patients who were only taking oral medication. Blood samples were extracted from patients who had begun IFX treatment to measure IFX trough concentration immediately before the fourth dose administration.

The pediatric Crohn’s disease activity index (PCDAI) ≤ 10 was used to define clinical remission (CR) as the outcome. Biochemical remission (BR) was defined as C-reactive protein (CRP) <0.5 mg/dL and fecal calprotectin <200 mg/kg [18]. Endoscopic remission (ER) was defined as a simple endoscopic score for Crohn’s disease (SES-CD) score of 0–2 [19]. The therapeutic range of IFX trough concentration was defined as ≥3 µg/mL [20]. Primary non-response was defined as all those patients who failed to respond to the standard induction doses (third dose) of infliximab. The primary non-responder was determined by the clinician by comprehensively evaluating the subjective symptoms, laboratory results, and fecal calprotectin.

The first aim of this study is to determine if there is a difference in CD severity or disease phenotype based on initial cytokines. The second aim of this study is to investigate if there is a difference in initial cytokines between patients who achieved remission after treatment and those who did not. Finally, the third aim of this study is to determine whether the initial cytokine affects treatment response in patients receiving IFX and whether it affects the trough concentration of IFX. This study’s methods were performed according to the relevant guidelines and regulations and were approved by the Clinical Research Ethics Committee of Samsung Medical Center. We followed appropriate written informed consent procedures for clinical data analysis. IRB File No.: SMC 2021-04-046.

### 2.2. Measurement of Cytokine Concentrations

The cytokine measurement included IL-6 (typically secreted by monocytes, macrophages, and dendritic cells), TNF-α (from macrophage, Th1, and NK cells), IL-17A (from Th17 and NK cells), and IL-10 (from Treg, Th2, monocytes, macrophages, and dendritic cells). The cytokine concentration was determined using the Luminex^®^ Performance Assay multiplex kit (R&D Systems, Minneapolis, MN, USA). Analyte-specific antibodies were precoated onto magnetic microparticles embedded with fluorophores in specific ratios for each microparticle region.

Pipetted microparticles, standards, and samples were pipetted into wells and immobilized antibodies bind the analytes of interest. After washing away any unbound substances, a biotinylated antibody cocktail specific to the analytes of interest was added to each well. Following the wash, streptavidin–phycoerythrin conjugate, which binds to the biotinylated antibody, was added to each well. After another wash, the microparticles were resuspended in buffer and analyzed with the Luminex^®^ MAGPIX^®^ Analyzer (Luminex Corporation, Austin, TX, USA).

### 2.3. Measurement of Infliximab Trough Concentrations and Anti-IFX Antibodies

The IDKmonitor^®^ Infliximab drug level enzyme-linked immunosorbent assay (ELISA) kit (IDKmonitor^®^, Bensheim, Germany) was used to measure the trough concentration. The bound-free IFX from the sample binds the plate’s specific monoclonal anti-IFX antibody. A washing step was performed, and a peroxidase-labeled antibody was added to remove all unbound substances. Tetramethylbenzidine is a peroxidase substrate. The yellow color’s intensity is proportional to the concentration of free IFX in the sample. The values obtained from the standard were used to generate a dose–response curve of the absorbance unit (optical density, OD) vs. concentration. This curve was used to directly calculate the concentrations of free IFX in the samples.

For the antibody detection, the IDKmonitor^®^ Infliximab free anti-drug (IFX) antibodies (ADA) ELISA kit (IDKmonitor^®^, Germany) was used. During sample preparation, the ADA were separated from the therapeutic antibody in order to acquire free ADA. By adding the peroxidase conjugate and the tracer, the unmarked therapeutic antibodies were replaced, and the marked antibodies can form a complex with the ADA. This was detected via the peroxidase conjugate with the peroxidase converting the substrate TMB to a blue product. The color was measured in a photometer at 450 nm. The interpretation was made using the cut-off control (10 AU/mL).

### 2.4. Statistical Analysis

Continuous variables were expressed as the median (interquartile range) for descriptive statistics, whereas categorical variables were expressed as absolute numbers (percentages). To analyze the average comparison between the two groups, Student’s T-test and the Mann–Whitney U test were used. Pearson correlation analysis was used to investigate correlation variables. The receiver operating characteristic (ROC) curve was used to assess the diagnostic ability. The diagnostic sensitivity, specificity, and area under the ROC curve were used to express the diagnostic performances. The cutoff values were determined using Youden’s J statistic. All of SPSS version 27 was used for all of the above statistical analyses (IBM Corporation, Armonk, NY, USA). A *p*-value of <0.05 was considered to be statistically significant.

## 3. Results

### 3.1. Patient Characteristics at Diagnosis

The median age of 30 patients was 13.7 years, and the majority were males, i.e., 80.0% males and 20.0% females (Table 1). There was a family history of IBD in six patients (20% of the total). The PCDAI score at diagnosis had a median value of 30.0. In the laboratory evaluation, median values of erythrocyte sedimentation rate (ESR) and CRP were elevated to 28 mm/h and 0.9 mg/dL, respectively, and median value of fecal calprotectin was also elevated at 1137.0 mg/kg. When Paris classification was used, the majority of patients (*n* = 23, 76.7%) had a diagnosis age of A1b (10 to <17 years).

During the location evaluation, the majority of patients (n = 25, 83.3%) were classified as L3 (ileocolic) type, and only two patients invaded L4a (proximal to Treitz ligament) upper gastrointestinal lesion. In the behavior evaluation, the majority of patients (n = 25, 83.3%) corresponded to the B1 (nonstricturing and nonpenetrating) type. Growth slowed in 13 (43.3%) of the patients. The median SES-CD value was 20.0. No patients had undergone abdominal surgery, such as bowel resection before diagnosis; however, 17 patients (56.7%) had undergone anal-related surgery.

### 3.2. Differences in Cytokines According to Disease Severity and Phenotypes

Laboratory results for hematocrit, albumin, ESR, CRP, and fecal calprotectin were examined to see they correlated with specific cytokines at the time of diagnosis. Pearson correlation analysis was used to assess the disease severity by investigating the correlations between the subjective factor PCDAI and the objective factor SES-CD score and cytokines (Appendix A). The only statistically significant finding was a correlation between IL-6 concentration and SES-CD score (*p* = 0.034). As the Pearson coefficient was positive at 0.388, the higher the IL-6 level, the higher the SES-CD score.

Disease locations were divided into compartments, such as invasion of the upper gastrointestinal tract, invasion of only the small intestine, invasion of only the large intestine, or invasion of the anus, and Pearson analysis was used to see if there were any relationships with cytokines; however, there were no statistically significant results.

### 3.3. Comparison of Clinical Characteristics of Patients Who Achieved ER and Those Who Did Not

As ER is the most accurate indicator for disease remission, and some patients achieve ER even after 4 months of treatment, the differences between those who achieved ER and those who did not were assessed (Table 2). Twenty patients received ER, whereas 10 did not. Patients who did not reach the ER had a statistically significantly lower body mass index and lower albumin levels due to severe diarrhea and weight loss (*p* = 0.047, <0.001, respectively) at the time of diagnosis.

As all the patients had moderate-to-severe CD, mesalazine and immunosuppressants were started from the start. However, one patient discontinued the mesalazine due to severe nausea, and another patient discontinued the azathioprine because of the severe side effects of hair loss. The majority of the patients in this study received anti-TNF-α treatment. Almost all (90%) patients in the group that achieved ER were treated with anti-TNF-α, and 80% of patients in the group that did not reach ER were treated with anti-TNF-α.

Nevertheless, there was a statistically significant difference in the IFX trough concentration between the two groups of patients who received anti-TNF-α treatment. Patients with ER had a median value of 6.1 µg/mL, whereas those without ER had a median value of 1.6 µg/mL, with a p-value of 0.010. Furthermore, the number of patients with an IFX range that was below the therapeutical range was significantly higher in the patients who do not have ER (50.0%).

After 4 months of treatment, fecal calprotectin, ESR, and SES-CD scores in the two groups were assessed as factors with statistically significant differences (*p* values = 0.005, 0.005, and <0.001, respectively).

### 3.4. Differences in Cytokines According to Remission in Patients Treated with Infliximab

The differences in initial cytokines between patients who achieved remission after treatment and those who did not were studied. At 4 months of treatment, the CR, BR, and ER were evaluated. As PCDAI includes many subjective factors, patients who met both CR and BR were counted (Table 3 and Appendix A). We divided 26 patients who received IFX treatment into two groups for those who achieved CR and BR (n = 18) and those who did not (n = 8) (Table 3). The TNF-α, IL-6, IL-10, and IL-17A levels in these two groups showed a statistically significant difference in this comparison (*p* = 0.027, 0.006, 0.017, and 0.032, respectively).

This finding indicates that patients who did not achieve CR or BR despite receiving infliximab treatment had significantly higher TNF-α concentrations than those who did. The comparison of cytokines between the two groups based on ER achievement yielded no statistically significant results (Appendix A). Appendix A depicts a box plot of the aforementioned results. All cytokines (TNF-α, IL-6, IL-10, and IL-17A) were significantly higher in patients who did not achieve BR or CR compared with those who did.

### 3.5. Relationships between Each Cytokine

Each cytokine was evaluated to see if there were positive or negative relationships with one another (Figure 2). IL-6, IL-10, IL-17A, and TNF-α were all found to have a statistically significant positive correlation. When examining the scatterplot, it was clear that all cytokines had a linear correlation with one another. The highest correlation was found between IL-17A and IL-10 with a Pearson correlation coefficient of 0.939, and a *p*-value of <0.001, followed by TNF-α and IL-10 with a Pearson correlation coefficient of 0.886 and a *p*-value of <0.001.

### 3.6. Relationship between Cytokine and Infliximab Trough Concentration and Anti-IFX Antibody Formation

We investigated whether initial cytokine concentrations were related to IFX trough concentrations and rapid anti-IFX antibody formation at the start of IFX maintenance treatment (Figure 3). Looking at the table in Figure 3, the cytokine that showed a statistically significant association with the IFX trough concentration before the fourth dose was TNF-α (*p* = 0.034). The Pearson coefficient is −0.425, indicating a negative correlation. This means that the higher the initial TNF-α concentration, the lower the IFX trough concentration with under therapeutic range (<3 µg/dL).

Figure 3 shows that natural log equation 3.149ln(x) + 14.674) is more appropriate than the linear equation (y = −3.149ln(x) + 14.674). There were seven patients with free anti-IFX antibodies measured even though blood was collected before the fourth dose. In patients with anti-IFX antibody formation, the trough concentration was measured to be less than 3 µg/mL. The relationship between this rapid antibody formation and initial cytokines did not obtain statistically significant results.

In this study, there were patients with an IFX trough concentration of less than 3 µg/mL, which did not satisfy the appropriate therapeutic concentration range and had a high initial TNF-α concentration. Consequently, the initial TNF-α cutoff value at which the IFX trough concentration could be in the therapeutic range was determined (Figure 4). The diagnostic capability of initial TNF-α concentration to predict under the therapeutic IFX trough concentration with ROC curve showed an area under the ROC of 0.730 and a *p*-value of 0.049. The TNF-α concentration cutoff value was 27.6 pg/mL at which the IFX trough concentration can be in the therapeutic range. The results show that if the initial TNF-α exceeds 27.6 pg/mL, the IFX trough concentration at the time of maintenance treatment may not reach the therapeutic range.

## 4. Discussion

We looked at cytokines produced during the inflammation process at the time of diagnosis in CD to see if there were any differences in disease phenotypes, severity, response to treatment, and effect on IFX tough concentration. If measuring the initial cytokines can predict the subsequent treatment response, measuring the concentrations can be useful in disease treatment. Although CD immunology is not fully understood, important cytokines have been identified [21]. We measured and analyzed the IL-6 (Typically secreted by monocytes, macrophages, and dendritic cells), TNF-α (from macrophage, Th1, and NK cells), IL-17A (from Th17 and NK cells), and IL-10 (from Treg, Th2, monocytes, macrophages, and dendritic cells).

The difference in cytokines according to the severity and phenotype of the disease at the time of diagnosis revealed that only initial IL-6 concentration was associated with SES-CD score, according to one of the study’s findings. We attempted to determine whether a large number of specific cytokines are associated with localization or phenotype, such as involvement of upper GI tract or anus at CD onset, as well as disease behaviors, such as penetrating or structuring.

Nevertheless, no statistically significant association was found with these disease phenotypes, and only the endoscopic SES-CD score, which can objectively evaluate disease severity, showed an association with a higher IL-6 value. There have been few studies on cytokines and CD disease phenotypes, and there have been no papers that evaluated cytokines in the pediatric patients. Kiernan, Miranda G., et al. [22] measured fibrocytes and cytokine in patients with CD and UC to see if there were any differences. Stallmach, Andreas, et al. [23] investigated whether elevated levels of cytokine transcripts (TNF-α, IFN-γ, CD40L, and IL-23) in tissues are associated with active CD.

It is difficult to distinguish the disease phenotype of CD-based on the current studies; however, it is determined that certain cytokines can be elevated in relation to disease severity, and IL-6 was associated with the initial disease severity in this study.

Our second major finding was that there was no difference in ER; however, it was confirmed that all cytokines were significantly lower in patients who achieved clinical and biochemical remission when the difference in cytokines was evaluated at 4 months of treatment. Our thoughts on this result are that the fourth-month treatment period is relatively short for evaluating ER with statistical significance.

Although there was no statistical significance, the concentrations of all cytokines were higher in patients who were unable to achieve the ER than in patients who were able to achieve the ER (Appendix A). We anticipate individuals that are classified as ER by colonoscopy at 1 year would more clearly show a relationship not clearly apparent at 4 months.

From this result, it can be seen that patients with a high cytokines burden may have a slow response to treatment; however, the intriguing and worth discussing point of result in this study is that all of the cytokines from each T cell subset are elevated together and have a positive linear relationship. Many biological agents are currently being developed to block various cytokines, and secukinumab, an anti-IL-17A, was also developed in line with this trend to prevent inflammation from the Th17 lineage [24]. Nevertheless, secukinumab has failed to demonstrate therapeutic efficacy in the treatment of CD, and a high rate of adverse events and increased disease severity compared with the placebo group were reported [25].

Based on the findings of our study, treating only IL-17A is ineffective because TNF-α and IL-6 levels are also elevated when IL-17A levels are elevated. Instead, by inhibiting the Th17 cell lineage, the homeostasis imbalance can further activate the Th1 or 2 cell lines, causing more TNF-α to be secreted. In support of this, there is also a report of a rapid onset of fulminant IBD following secukinumab infusion [26].

IL-17A has recently emerged as a protective cytokine for inflammatory bowel diseases, as a result of which clinical trials with secukinumab or brodalumab in patients with IBD resulted in enhanced Candida infections and increased intestinal inflammation. Although IL-17A can drive inflammation that damages the gut mucosa, IL-17A also play protective roles in limiting fungal and bacterial infection of the gut [27]. Elevation of IL-17A as a gut protective role, such as IL-10 discussed below, can be thought of as a result of the host’s immune response process to alleviate the high burden of inflammation.

IL-10 is a regulatory cytokine that inhibits both antigen presentation and the subsequent proinflammatory release of cytokine [14]. Consequently, we expected the IL-10 concentration to be relatively low when the TNF-α was high in patients with severe inflammation prior to analysis but IL-10 also yielded high results. We also divided IL-10 by TNF-α to see if this value was relatively low in patients with severe disease; however, it was not statistically significant.

According to our observations, cytokines during the inflammation process, including all T cell subsets, appear to be equally activated in response to inflammatory stimulation. Since IL-10 is also secreted from monocytes, macrophage, and dendritic cells involved in the inflammatory process, it is thought to be measured highly when the inflammatory response is high. It can also be thought of as increasing the secretion of IL-10 in Treg to alleviate the inflammation. In this context, it is thought that the IL-10 concentration of nonresponders with a higher burden of inflammation was higher than that of responders in Table 3 and Appendix A of our results.

Our final major finding was that when we looked at the relationship between cytokines and IFX trough concentrations, patients with high initial TNF-α concentrations had lower IFX trough concentrations, which were below the appropriate treatment range. There are many previous studies [28,29] that covered on the therapeutic effect of IFX based on trough concentration that results in therapeutic range of the IFX.

A recent review article published by Shmais et al. [30] mentions the importance of proactive therapeutic drug monitoring of anti TNF-α for determining escalation or de-escalation treatment in patients using anti TNF-α. Further in this field, we were curious about how the initial cytokines differed in patients with primary nonresponse to IFX versus patients who required rapid dose intensification. Knowing differences of the initial cytokine concentrations greatly improves one’s ability to predict the treatment response.

In a previous study, Billiet, Thomas, et al. [31] found that serum IFX trough concentration increased more significantly in responders than nonresponders after each infliximab infusion—that is, primary nonresponders had low trough concentrations even after IFX administration; however, they had not measured or evaluated the initial TNF-α at the time of diagnosis.

In this study, we confirmed that the trough concentration of IFX was below the treatment range when the initial TNF-α was high. These patients had a high rate of anti-IFX antibody formation, and dose intensification was used during treatment as in previous studies [32,33,34]. In two recent studies involving pediatric patients on anti-IFX antibody production after starting infliximab treatment, the anti-IFX antibody production rates were 14.7% and 31%. In that study, the anti-IFX antibody production rate was 23.3%, which did not deviate significantly from the anti-IFX antibody production rate confirmed in the previous two studies [35,36].

Trotta, Maria Consiglia, et al. [36] revealed that the time of anti-IFX antibody formation was earlier in children and especially in female, and asserted that anti-IFX antibodies can be produced from 3 months of using infliximab as in our study. In this study, seven patients rapidly formed anti-IFX antibodies after receiving only three doses of infliximab, and it is necessary to evaluate whether the rapid anti-IFX antibody formation is related to high initial TNF-α levels with a larger number of samples.

In patients with high initial TNF-α concentration, it appears that anti-TNF-α (IFX) concentrations are rapidly consumed by the drug binds to TNF-α. Consequently, we believe that patients with high initial TNF-α concentrations should be given a high dose of infliximab to stay within the known therapeutic range and avoid anti-IFX antibody formation. The cutoff value for considering administering a higher dose of infliximab in this study was 27.6 pg/mL.

Our findings suggest that measuring cytokines at the time of diagnosis can be used to predict treatment response. When measuring multiple cytokines during the inflammation process, including each T cell subset, if one cytokine burden is high, the other cytokines are also high. The treatment response to IFX can be predicted if the initial TNF-α concentration of pediatric patients who are scheduled for IFX treatment is measured, the treatment response to IFX can be predicted.

Furthermore, when the initial TNF-α concentration is high, giving a 1.5-fold (7.5 mg/kg) or two-fold (10 mg/kg) higher dose of IFX rather than routinely administering 5 mg/kg of IFX may be more appropriate to maintain an appropriate therapeutic concentration. It may also be considered to actively shorten the dosing interval to 6 or 4 weeks instead of 8 weeks. The main limitation of this study is that it was conducted on a small number of patients because it was a prospective study with pediatric patients.

Consequently, there is a change that the statistical significance was underestimated. Furthermore, the ROC curve is not smooth, and the AUROC is 0.73, which is considered acceptable because the sample size of this study is small. Based on this result, it can be expected to have a higher predictive value when the sample size is increased but re-evaluation is required. In addition, as this study is a short follow-up study, it is necessary to evaluate the long-term response of more than 1 year in the future.

Furthermore, peripheral blood cytokine levels were measured but the tissue cytokine levels were not measured. The cytokine level measured in tissue may show more accurate results than measured in peripheral blood. However, it is meaningful to measure and evaluate the peripheral blood cytokine levels because there is a relationship between the tissue cytokine levels and blood cytokine levels [37], and measuring the tissue cytokine levels is difficult to apply clinically. This study on the relationship between the initial TNF-α concentration and the IFX concentration was the first study that had ever been conducted in the adult studies, and we think that this study will play an important role as a preliminary study that may change the treatment regimen in the future.

## 5. Conclusions

Measuring cytokines at the time of diagnosis can be used to evaluate the disease activity and to predict the response after treatment. According to the severity associated with the SES-CD score, IL-6 had a linear relationship. When multiple cytokines during the inflammation process, including each T cell subset, were measured, if one cytokine burden was high, the other cytokines were also high. In pediatric patients, measuring the initial TNF-α concentration can predict the treatment response to IFX. When the initial TNF-α concentration is greater than 27.6 pg/mL, giving a higher dose of IFX rather than routinely administering 5 mg/kg of IFX may be more appropriate to maintain an appropriate therapeutic concentration (≥3 µg/mL).

## Figures and Tables

**Figure 1 biomedicines-10-02372-f001:**
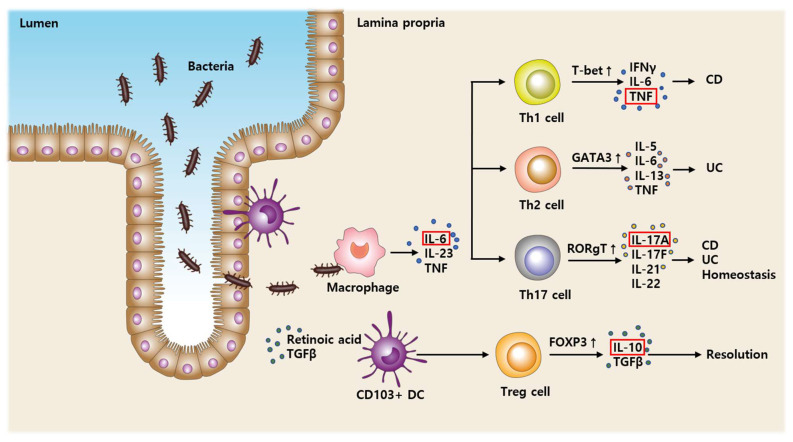
Depicting the process of T cells activation using cytokines from each T cell subset. IL: Interleukin, DC: Dendritic cell, TGF: Transforming growth factor, Th: T helper, Treg: T regulatory, T-bet: T-box expressed in T cells, GATA3: transcription factor that, in humans, is encoded by the GATA3 gene, RORγt: RAR-related orphan receptor gamma, FOXP3: Forkhead box P3, TNF: Tumor necrosis factor, IFN: Interferon, CD: Crohn’s disease, and UC: Ulcerative colitis.

**Figure 2 biomedicines-10-02372-f002:**
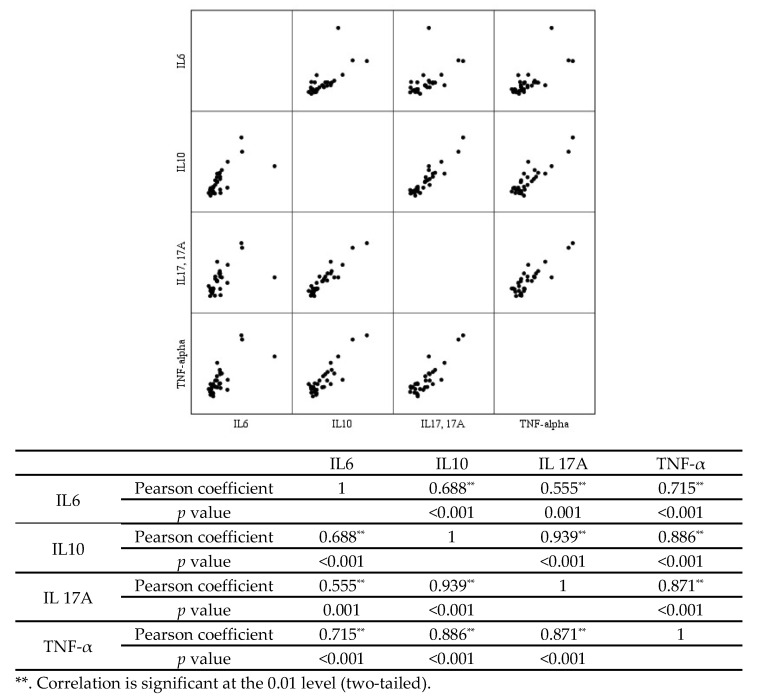
Correlation between each cytokine evaluated by scatterplot and Pearson correlation analysis. IL; interleukin, and TNF; tumor necrosis factor.

**Figure 3 biomedicines-10-02372-f003:**
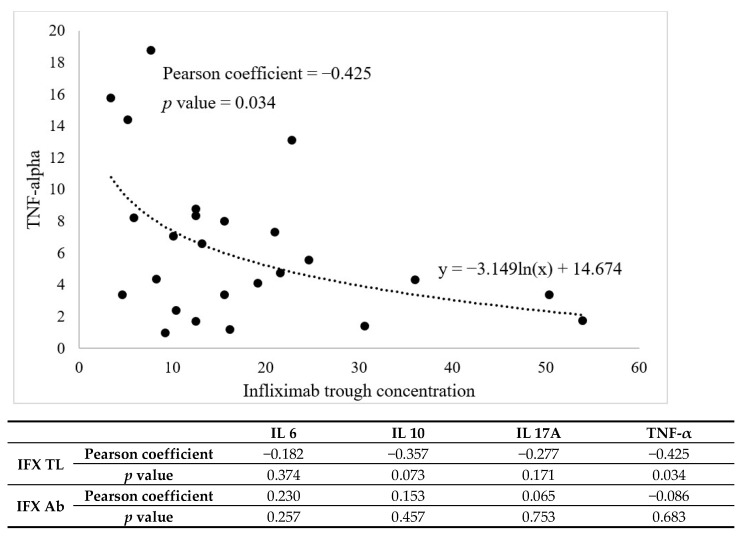
Pearson correlation analysis was used to assess the relationship between each cytokine and infliximab trough concentration, as well as anti-IFX antibody concentration, prior to the fourth dose administration.

**Figure 4 biomedicines-10-02372-f004:**
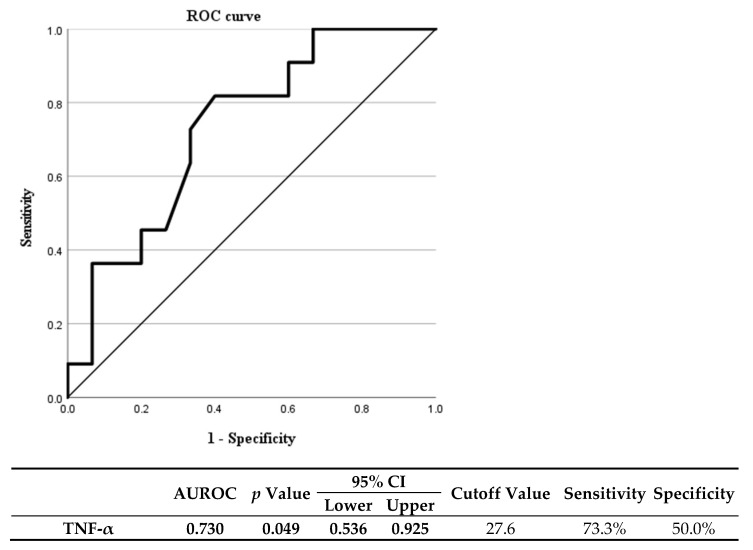
Diagnostic capability of initial TNF-α concentration with receiver operating characteristics curve in the patients whose infliximab trough concentration measured before the fourth administration was less than 3 µg/mL. TNF-α; Tumor necrosis factor-α, and AUAROC; Area under receiver operating characteristic.

**Table 1 biomedicines-10-02372-t001:** Baseline patient characteristics at the initial diagnosis of Crohn’s disease.

Characteristics		Number of Patients = 30
Age, years		13.7 (12.2–16.1)
Sex, M/F		24 (80.0)/6 (20.0)
BMI, kg/m^2^		18.0 (15.9–21.1)
PCDAI ^a^		30.0 (25.0–35.0)
Family history of IBD (Y)		6 (20.0)
Hematocrit, %		37.8 (35.1–41.9)
Albumin, g/dL		4.1 (3.6–4.3)
ESR, mm/h		28.0 (18.0–72.3)
CRP, mg/dL		0.9 (0.3–2.0)
Fecal calprotectin, mg/kg		1137.0 (1000.0–2543.3)
Paris classification at diagnosis		
Age at diagnosis	A1a	0
	A1b	23 (76.7)
	A2	7 (23.3)
Location	L1	4 (13.3)
	L2	1 (3.3)
	L3	25 (83.3)
	L4a	2 (6.6)
	L4b	0
Behavior	B1	25 (83.3)
	B2	4 (13.3)
	B3	1 (3.3)
	B2B3	0
	p	22 (73.3)
Growth	G0	17 (56.7)
	G1	13 (43.3)
SES-CD ^b^ score		20.0 (14.3–27.0)
ANCA ^c^-positive (Y)		15 (50.0)
ASCA ^d^-positive (Y)		14 (46.7)
History of IBD-related surgeries Fistulotomy or Seton placement (Y) Laparoscopy or Bowel resection (Y)		17 (56.7) 0

Values are n (percentage) or median (interquartile range); Y, yes. ^a^ The pediatric Crohn’s disease activity index (PCDAI) score can range from 0 to 100, with higher scores signifying more active disease. A score of <10 is consistent with inactive disease, 11–30 indicates mild disease, and >30 is moderate-to-severe disease. ^b^ The simple endoscopic score for Crohn’s disease (SES-CD) assesses the size of mucosal ulcers, the ulcerated surface, the endoscopic extension, and the presence of stenosis. ^c^ Antineutrophil cytoplasmic antibodies. ^d^ Anti-Saccharomyces cerevisiae antibody.

**Table 2 biomedicines-10-02372-t002:** At 4 months after treatment, a comparison of initial clinical characteristics and clinical improvement between patients with and without endoscopic remission was made. ER; Endoscopic remission.

Characteristics	ER (N = 20)	No ER (N = 10)	*p* Value
Initial clinical characteristics at diagnosis			
Age, years	13.7 (12.5–16.7)	13.3 (12.2–14.6)	0.169 ^e^
Sex, M/F	16 (80.0)/4 (20.0)	8 (80.0)/2 (20.0)	1.000 ^f^
BMI, kg/m^2^	18.5 (17.3–21.5)	16.3 (15.6–19.1)	**0.047** ^e^
PCDAI ^a^	30.0 (25.0–32.5)	35.0 (30.0–37.5)	0.380 ^e^
Fecal calprotectin, mg/kg	1189.0 (976.3–2253.8)	1327.0 (1000.0–2854.0)	0.302 ^e^
Hematocrit, %	38.2 (36.5–42.3)	37.3 (32.7–39.5)	0.098 ^e^
Albumin, g/dL	4.3 (4.1–4.3)	3.6 (3.4–3.8)	**<0.001** ^e^
ESR, mm/h	25.5 (17.5–37.5)	50.0 (19.5–80.8)	0.266 ^e^
CRP, mg/dL	0.7 (0.3–1.2)	1.7 (0.8–2.7)	0.292 ^e^
SES-CD ^b^ score	19.5 (8.8–26.3)	21.5 (15.8–27.8)	0.532 ^e^
Family history of IBD (Y)	5 (25.0)	1 (10.0)	0.298 ^f^
ANCA ^c^-positive (Y)	8 (40.0)	7 (70.0)	0.130 ^f^
ASCA ^d^-positive (Y)	7 (35.0)	7 (70.0)	0.222 ^f^
Treatment			
EEN (Y) Mesalazine (Y) Immunosuppressants (Y) Corticosteroid (Y)	12 (60.0) 19 (95.0) 20 (100) 1 (5.0)	8 (80.0) 10 (100) 9 (90.0) 0	0.289 ^f^ 0.264 ^f^ 0.330 ^f^ 0.330^f^
Anti-TNF α (Y) Trough concentration Trough concentration <3 ug/mL (Y)	18 (90.0) 6.1(3.9–8.4) 2 (11.1)	8 (80.0) 1.6 (1.2–3.7) 4 (50.0)	0.516 ^f^ **0.010** ^e^ **<0.001** ^f^
Clinical characteristics at 4 months after treatment			
PCDAI ^a^	5.0 (0–5.0)	5.0 (1.3–10.0)	0.314 ^e^
Fecal calprotectin, mg/kg	68.5 (45.8–157.3)	725.5 (395.3–1187.8)	**0.005** ^e^
Hematocrit, %	42.2 (40.3–44.0)	41.3 (38.2–43.3)	0.495 ^e^
Albumin, g/dL	4.7 (4.4–4.7)	4.5 (4.3–4.7)	0.169 ^e^
ESR, mm/h	2.5 (2.0–4.0)	9.5 (3.0–19.5)	**0.005** ^e^
CRP, mg/dL	0.05 (0.03–0.06)	0.07 (0.04–0.10)	0.058 ^e^
SES-CD score	0.0 (0.0–0.0)	9.0 (4.5–13.5)	**<0.001** ^e^

Values are n (percentage) or median (interquartile range); Y, yes. ^a^ The pediatric Crohn’s disease activity index (PCDAI) score can range from 0 to 100, with higher scores signifying more active disease. A score of <10 is consistent with inactive disease, 11–30 indicates mild disease, and >30 is moderate-to-severe disease. ^b^ The simple endoscopic score for Crohn’s disease (SES-CD) assesses the size of mucosal ulcers, the ulcerated surface, the endoscopic extension and the presence of stenosis. ^c^ Antineutrophil cytoplasmic antibodies. ^d^ Anti-Saccharomyces cerevisiae antibody. ^e^ T-test. ^f^ Mann–Whitney test.

**Table 3 biomedicines-10-02372-t003:** Comparison of the initial cytokine profiles between patients with and without clinical remission and biochemical remission among patients treated with anti-TNF-α. ER; Endoscopic remission, SD; Standard deviation, TNF; Tumor necrosis factor, CR; Clinical remission, and BR; Biochemical remission.

	Infliximab Use + CR + BR (N = 20)		Infliximab Use + No CR or BR (N = 6)		*p* Value
	Mean	SD	Mean	SD	
TNF-alpha, pg/mL	14.6	10.7	27.9	16.0	**0.027** ^a^
Interleukin 6, pg/mL	18.4	13.7	48.2	38.7	**0.006** ^a^
Interleukin 10, pg/mL	93.9	76.1	202.2	132.3	**0.017** ^a^
Interleukin 17A, pg/mL	8.1	6.6	16.0	10.0	**0.032** ^a^

^a^ T-test.

## Data Availability

All data generated or analyzed during this study are included in this published article.

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
