# Peer review of "Cytokine Profile at Diagnosis Affecting Trough Concentration of Infliximab in Pediatric Crohn’s Disease"

_biomedicines, 2022, doi:10.3390/biomedicines10102372_

Round 1

Reviewer 1 Report

Kwon et al suggest in their paper that measuring cytokine serum levels at diagnosis may predict trough concentration of infliximab (IFX) in pediatric Crohn’s disease and, hence, treatment response. Specifically, they conclude that patients with serum TNF-α levels greater than 27.6 pg/mL at treatment start may not obtain disease remission with standard IFX doses (5 mg/kg), thus possibly requiring higher drug amounts.

The major concern with the conclusions of this paper is the lack of serum anti-infliximab antibody measurement in the patient population. Infliximab is a chimeric monoclonal antibody whose murine sequences are known to induce neutralizing anti-IFX antibodies, causing IFX trough concentrations to be low. Without this piece of information, it remains unknown whether low trough IFX concentrations are truly dependent on high baseline serum TNF-a concentrations or anti-IFX antibodies.

Further points:

-          Please change RORrt to RORgT in the image of figure 1.

-          Page 4, lines 163-165: “In the laboratory evaluation, erythrocyte sedimentation rate (ESR) and CRP were elevated to 28 mm/h and 0.9 mg/dL, respectively, and fecal calprotectin was also elevated to 1137.0 mg/kg”. What do these values refer to? I guess they are still medians, but this is not specified.

-          Table 1: again, I guess the values reported correspond to medians, but there’s no specification.

-          Table 1: define (Y), ANCA, ASCA, etc., as done for Table 2.

-          Page 7, lines 208-209: “Although the majority of the patients received anti-TNF-α treatment, the proportion of patients who achieved ER was higher than those who did not (90.0% vs. 80.0%)…” It is unclear what the percentages refer to, the sum exceeds 100%. I guess the Authors intended to compare the rates of endoscopic remission between infliximab-treated and non-infliximab-treated patients. If so, the statement should be rephrased.

-          Page 9, lines 263-264, the Authors state “In this study, there were patients with an IFX concentration of <3 μg/mL, which was within the therapeutic range…”. The therapeutic range, as stated in the Methods section (page 3, line 111), was ³3 mg/mL. Please correct.

-          Page 8, line 253: please delete “4” in Figure 34.

-          The receiver operating characteristics (ROC) curve results appear not to be enough robust to allow reliable predictions. The Authors should comment on this point.

-          Endoscopic remission (ER) is clearly a more reliable outcome than clinical or biochemical remission. According to Supplementary Table 2, there were no differences in serum cytokine concentration at baseline between patients who obtained ER and those who didn’t. However, differences were noted when considering clinical and biochemical remission. How would the Authors explain this apparent discrepancy?

Reviewer 2 Report

This is a paper looking at measures of cytokines to predict treatment response. It is well written and interesting to the topic of IBD. 

unfortunately the sample size is small and that could have played a role in the results. it would have been interesting to obtain tissue samples although that would not be easy.

when you mention higher dose do you mean raising from 5 to 7.5 mg/kg or shortening the interval. It is interesting to look at shortening the interval as well. 

Reviewer 3 Report

This is a timely manuscript, addressing a very important topic "Pediatric Crohn's disease". There are some issues that I would like to see clarified, as well as some editing of the English. 

1-Background Line 10, "during (add "the") inflammation process... .  Line 11, insert  "of pediatric crohn's disease" after "in response to treatment".......  This sentence. In general, this,is a long statement and there should be on or two "commas"  Lines 21-22 "space between therapeutic and IFX". This sentence, line 21-22, is a little vague (e.g. "to predict under therapeutic IFX trough concentration..... is a little vague and should be reconstructed. Line 54 "each T cell should be "each T cell subset". 

Figure 1 Very good figure, my question, are these results obtained from the authors work, or from the literature. Should be noted. Lines 116-120 two aims of the study "it would be clearer if these to aims were separated , e.g Aim 1 ...... .  Aim 2:....... . 

Tables 1 and 2 - Good tables, but great deal of information. My suggestion, particularly for a novice reading this paper, is to have a separate sheet  which describes some terms, e.g. ESR, CRP, PCDAI, etc.

Line 252-253, "looking at the table in Figure 34", should be Figure 3. Page 279 "produced during (add "the") inflammation.......  page 284-285 "cella like macrophage" ?

Throughout this manuscript, the authors refer to T cells in the singular, e.g.line 50 " The gut has T cell classified"  Should be "T cells. In the majority they should be referred to as "T cells" and not "T cell". Or "T cell subset".

In a brief literature search, I found lest 4 manuscripts ( one being a revue) that have relevance to the current manuscript: " Inflammatory Intest Dis 2022;7:50-58;  BMC Gastrointerol ( 2021) 21:77;  Journal of Crohn"s and Colitis , 2019, 189-197;  JPGN Vol 64, No.2, Feb 2017. " At least some should be included in the references."

In summary, I believe the current manuscript has Very Good merit to be published, but it needs to be significantly revised. 

Reviewer 4 Report

This is a very interesting and clinically very important research. Several issues should be resolved as follows: 

1.The authors described that the source of IL-10 is Treg. However, IL-10 is secreted by many sorts of cells such as Th2, monocyte/macrophages, dendritic cells, epithelial cells, etc. The description should be corrected.

2.TNF-a is also secreted by many sorts of cells besides macrophages or Th1 cells. The description should be corrected.

3.IL-17A is now considered as a protective cytokine for inflammatory bowel diseases, necessary for protecting intestinal barrier. Rather IL-17F is considered as a disease-aggravating cytokine, inhibiting the development of Treg. These issues should be incorporated into Discussion.

4.How  do the authors think of positive correlation of IL-10 concentration  with other inflammatory cytokines, and of increased IL-10 concentrations in non-responders than in responders? Do these indicate counter-regulatory effects in inflammatory conditions in non-responders? Since authors insist Treg as IL-10 source, do these results mean the increased Treg activities in non-responders compared to responders? These issues should be discussed.

Round 2

Reviewer 1 Report

Response to major concern:

The Authors state they measured IFX antibodies, specifically, they affirm “The contents below are some parts in the first manuscript we wrote about antibody.

Measurement of infliximab antibodies

For the antibody detection, a commercial ELISA kit (IDKmonitor®, Germany) was used. During sample preparation, the anti-drug antibodies (ADA) are separated from the therapeutic antibody in order to acquire free ADA. By adding the peroxidase conjugate and the tracer, the unmarked therapeutic antibodies are replaced and the marked antibodies can form a complex with the ADA. It is detected via the peroxidase conjugate with the peroxidase converting the substrate TMB to a blue product. The colour is measured in a photometer at 450nm. The interpretation is made using the cut-off control (10 AU/mL).”

I checked again the old version of the manuscript, but didn’t find this paragraph. The only paragraphs I see are “Measurement of cytokine concentrations” and “Measurement of infliximab trough concentrations “. Nevertheless, if 7 patients turned out to be positive for anti-IFX antibodies, as the Authors state in their response to the issues raised during peer review, were they excluded from the subsequent statistical analyses? In the previous manuscript version, the Authors agree that the receiver operating characteristics (ROC) curve results appeared not to be enough robust to allow reliable predictions; by excluding 7 patients developing anti-IFX antibodies out of 30, I doubt the results would still be significant.  Moreover, figure 4 legend reads “Diagnostic capability of initial TNF-α concentration to predict under therapeutic IFX trough concentration with receiver operating characteristic curve in the patients whose infliximab trough concentration measured before the fourth administration was 3 µg/mL or higher”. If subtherapeutic IFX concentrations are lower than 3 mcg/ml, the analysis should have been performed on patients whose infliximab trough concentration measured before the fourth administration was 3 µg/mL or lower, rather than higher. IFX concentrations higher than 3 mcg/ml are actually considered therapeutically ok by the Authors, as stated in the Methods section.
